# The Antioxidant Effects of Trypsin-Hydrolysate Derived from Abalone Viscera and Fishery By-Products, and the Angiotensin-I Converting Enzyme (ACE) Inhibitory Activity of Its Purified Bioactive Peptides

**DOI:** 10.3390/md22100461

**Published:** 2024-10-07

**Authors:** Jun-Ho Heo, Eun-A Kim, Nalae Kang, Seong-Yeong Heo, Ginnae Ahn, Soo-Jin Heo

**Affiliations:** 1Jeju Bio Research Center, Korea Institute of Ocean Science and Technology (KIOST), Jeju 63349, Republic of Korea; unknown0713@kiost.ac.kr (J.-H.H.); euna0718@kiost.ac.kr (E.-A.K.); nalae1207@kiost.ac.kr (N.K.); syheo@kiost.ac.kr (S.-Y.H.); 2Department of Marine Biology, University of Science and Technology, Daejeon 34113, Republic of Korea; 3Department of Food Technology and Nutrition, Chonnam National University, Yeosu 59626, Republic of Korea; gnahn@jnu.ac.kr

**Keywords:** abalone, viscera, by-product, peptides, antioxidant, anti-hypertensive

## Abstract

Abalone is a rich source of nutrition, the viscera of which are discarded as by-product during processing. This study explored the biological activities of peptides derived from abalone viscera (AV). Trypsin-hydrolysate of AV (TAV) was purified into three fractions using a Sephadex G-10 column. Nine bioactive peptides (VAR, NYER, LGPY, VTPGLQY, QFPVGR, LGEW, QLQFPVGR, LDW, and NLGEW) derived from TAV-F2 were sequenced. LGPY, VTPGLQY, LGEW, LDW, and NLGEW exhibited antioxidant properties, with IC_50_ values of 0.213, 0.297, 0.289, 0.363, and 0.303 mg/mL, respectively. In vitro analysis determined that the peptides VAR, NYER, VTPGLQY, QFPVGR, LGEW, QLQFPVGR, and NLGEW inhibited ACE, with IC_50_ values of 0.104, 0.107, 0.023, 0.023, 0.165, 0.004, and 0.146 mg/mL, respectively. The binding interactions of ACE-bioactive peptide complexes were investigated using docking analysis with the ZDCOK server. VTPGLQT interacted with HIS513 and TYR523, and QLQFPVGR interacted with HIS353, ALA354, GLU384, HIS513, and TYR523, contributing to the inhibition of ACE activity. They also interacted with amino acids that contribute to stability by binding to zinc ions. QFPVGR may form complexes with ACE surface sites, suggesting indirect inhibition. These results indicate that AV is a potential source of bioactive peptides with dual antioxidant and anti-hypertensive dual effects.

## 1. Introduction

Bioactive peptide consists of 2–20 amino acids that are important for regulating the essential activities of human metabolism [1,2]. Bioactive peptides have various activities, such as antioxidant, anti-inflammatory, and anti-hypertension activities, and skin improvement [3,4,5,6]. Food-derived bioactive peptides are safe and have health values [7]. Seafood is a strategic source of bioactive peptides and has, therefore, attracted research attention [8].

Reactive oxygen species (ROS) are physiological metabolites that cause oxidative damage to macromolecules in cells and are eliminated by the antioxidant defense system. Hydrogen peroxide (H_2_O_2_), a representative ROS, interacts with the peroxidase enzyme in humans [9]. Recent studies have reported that bioactive peptides with H_2_O_2_ resistance exhibit dual anti-hypertensive properties [10]. Hypertension is a cardiovascular condition that affects 30–45% of the population (2023) [11]. The World Health Organization (WHO) reported that hypertension is the most important cause of death [12]. Angiotensin-I converting enzyme (ACE) inhibition is an effective anti-hypertensive therapy.

Previous studies have reported that bioactive peptides isolated from proteins supplied by digestion disturb ACE [13]. Marine peptides derived from fish, jellyfish, sea cucumber, sponge, and oysters demonstrate excellent anti-hypertensive properties [14]. Indeed, previous studies have reported the dual effects of antioxidant and anti-hypertensive bioactive peptides derived from marine resources such as seaweeds and marine diatoms; however, detailed data remain lacking [15,16,17].

The sea has enormous diversity, making it an important source of novel compounds. As marine consumption increases, fishery by-products are produced during processing [18]. These by-products are difficult to dispose of owing to their rapidly decomposing vulnerabilities and are accompanied by environmental and economic problems [19,20]. Marine molluscs include 48,584 species and are a rich source of proteins [21,22,23]. Abalone is a seafood consisting of gastropod molluscs which are highly produced and consumed as the dominant source of nutrition, especially in Asia [24]. Abalone viscera (AV) account for 15–25% of the body weight and cannot be disposed of directly due to their high organic content [25,26]. With industrial development, AV are drawing attention as fishery by-products, with the possibility of becoming affordable, economical, and high-value-added products [27]. Viscera contain abundant proteins that serve as sources of bioactive peptides [28]. Enzymatic hydrolysis is the most beneficial and frequently used method for extracting bioactive peptides from materials [29].

In this study, to illustrate fishery by-products as a source of protein for bioactive peptides, the dual AV antioxidant and anti-hypertensive effects and the interaction with ACE are confirmed through docking analysis [30,31].

## 2. Results and Discussion

### 2.1. Proximate Compositions of TAV

The proximate compositions of the trypsin-hydrolysate of AV (TAV) are summarized in Table 1. The yield was 75.44 ± 1.22%, calculated by subtracting the dry weight of the residue from the TAV and expressed as a percentage. The protein, polysaccharide, and total polyphenol contents of the TAV were 48.19 ± 0.58, 6.30 ± 0.26, and 2.36 ± 0.03%, respectively. TAV’s primary chemical component is protein; hence, it was selected. The yield of enzymatic hydrolysate assisted by another protease (Alcalase, Flavourzyme, Neutrase, and Protamex) of AV previously reported was 28.02–34.64%, with a protein content of 19.30% [32]. Therefore, compared with the results of this study, trypsin-hydrolysis can be adopted as an efficient procedure to obtain a high protein yield by achieving a content difference of approximately double.

### 2.2. Separation of Peptides from TAV and Determination of Their H_2_O_2_ Radical Scavenging Activity

Enzyme hydrolysates purified according to molecular size exhibit antioxidant properties as the molecular weight decreases [33]. Therefore, we conducted purification to investigate the antioxidant properties of the TAV according to molecular size. The TAV was purified using GFC on a Sephadex G-10 column to separate the active peptides. As shown in Figure 1, the TAV was separated into three fractions according to molecular size. When the highest concentration (4 mg/mL) of all fractions was treated, the TAV and TAV-F1 showed H_2_O_2_ scavenging effects close to 90%, and TAV-F2 and TAV-F3 showed scavenging effects close to 100%. The IC_50_ value of the scavenging effects of the TAV was 1.34 ± 0.01 mg/mL, which was not significantly different from that of TAV-F1 of 1.37 ± 0.19 mg/mL. However, the IC_50_ values of TAV-F2 and TAV-F3 were 0.41 ± 0.08 and 0.47 ± 0.04 mg/mL, respectively, which were significantly low (Figure 2).

In a previous study, the DPPH and hydroxyl scavenging effects of the TAV showed IC_50_ values of 4 and 23 mg/mL, respectively [34]. Conversely, this study investigated the TAV at a significantly lower concentration for H_2_O_2_ scavenging activity. In addition, the antioxidant properties of the low-molecular-weight peptides were determined by measuring the IC_50_ values of TAV-F2 and TAV-F3, which were significantly lower than that of the TAV.

### 2.3. Identification of Separated Active Peptides and Determination of Their H_2_O_2_ Radical Scavenging Activity

The efficacy of the separated peptides was evaluated to identify bioactive peptides with antioxidant properties. TAV-F2, which was preferred, considering its effects and yield, was analyzed using a Micro Q-TOF mass spectrometer as a fraction of the amino acid sequence. Nine main peaks were detected in the chromatogram (Figure 3), which corresponded to the following peptides: VAR, NYER, LGPY, VTPGLQY, QFPVGR, LGEW, QLQFPVGR, LDW, and NLGEW. To compare the efficacy of the fraction, peptides with the same sequence were synthesized to measure H_2_O_2_ scavenging activity (Figure 4 and Table 2). The H_2_O_2_ scavenging activity of the LGPY was 0.213 ± 0.010 mg/mL, which was the lowest IC_50_ value, and VAR, QFPVGR, and QLQFPVGR activity were not detected. The IC_50_ value of NYER exceeded 0.4 mg/mL, while the IC_50_ values of VTPGLQY, LGEW, LDW, and NLGEW were 0.297 ± 0.016, 0.289 ± 0.012, 0.363 ± 0.020, and 0.303 ± 0.001 mg/mL, respectively, which were lower than the IC_50_ value of TAV-F2. Previous studies have shown that the presence of tryptophan in functional peptides is closely related to antioxidant efficacy [35]. Additionally, C-terminus residues contribute to the improvement in the antioxidant ability of peptides containing aromatic amino acid Tyrosine, and the presence of Leucine in the N-terminus has also been reported to have antioxidant efficacy [36]. The average concentration of AV-derived bioactive peptides isolated using different protocols has been reported to be 0.611 mg/mL [37]. The bioactive peptides derived from TAV-F2 presented a clear scavenging effect and may serve as a factor that allows for the utilization of the TAV as an antioxidant source for bioactive peptides from fishery by-products [38,39].

### 2.4. In Vitro Analysis of Active Peptides on ACE Inhibition

Recently, materials that possess both antioxidant and anti-hypertensive properties have been designed for the functional food industry, focusing on bioactive peptides [40]. Previous studies have revealed the ACE-inhibitory activity of bioactive peptides, including their antioxidant effects [10]. Marine gastropods have also been studied as the sources of the dual effects [41]. However, only limited attempts have been made to use marine resources for this purpose, and there has been no access to AV, a by-product of fisheries. Marine products are receiving increasing attention as functional materials which have the effects of anti-hypertension, and the proteins derived from them are important sources of physiological activities [42]. Research on ACE inhibition has demonstrated that peptides capable of inhibiting ACE are short-sequenced [43,44]. Various studies have been conducted on ACE inhibition by bioactive peptides derived from hydrolysates of fishery by-product enzymes obtained using proteases such as trypsin [45].

Therefore, this study evaluated the ACE-inhibitory effects of bioactive peptides derived from TAV-F2 through in vitro analysis to investigate their dual effects. The ACE inhibition activity IC_50_ values are listed in Table 3. Except for LGPY and LDW, the peptides of IC_50_ values were VAR (0.104 ± 0.010), NYER (0.107 ± 0.004), LGEW (0.165 ± 0.011), and NLGEW (0.146 ± 0.009 mg/mL). QLQFPVGR showed the best efficacy with a value of 0.004 ± 0.001 mg/mL, followed by VTPGLQY and QFPVGR at 0.023 ± 0.001 and 0.023 ± 0.003 mg/mL, respectively. The ACE inhibitory efficacy of the isolated seahorse-derived bioactive peptides was measured via comparable protocols for IC_50_ values of 0.088–0.171 mg/mL in an in vitro study [40]. Through comparison with these results, the inhibition efficacy of QLQFPVGR, VTPGLQY, and QFPVGR was confirmed.

### 2.5. In Silico Analysis of Active Peptides on ACE Inhibition

Computer prediction simulations were used to investigate the interactions between the bioactive peptides derived from TAV-F2 and ACE. ZDOCK is an automated simulation that selects poses by clustering according to ligand position, and has a high level of understanding in the modeling and analysis of protein–protein complexes essential for physiological activity [46,47,48]. The ZRANK score, which showed an accurate hit rate, was adopted and its efficacy was assessed [49]. Automated docking analysis was used to predict the binding interaction between peptides with short amino acid sequences and the target protein (Appendix A). The positive control group (captopril) used in vitro is a macromolecule compound that inhibits it as an ACE target and has not been studied using ZDOCK, which focuses on protein–protein binding in this paper. In particular, QLQFPVGR, VTPGLQY, and QFPVGR were evaluated as having the lowest ZRANK scores of −82.468, −81.500, and −78.016, respectively, whereas the peptides, except for VAR, had scores lower than −50.000 (Table 3). The in vitro and in silico results for the three peptides with the best efficacy were equivalent. The docking poses are shown as 2D diagrams and 3D crystalline structures to investigate the interaction of the combined docking complexes (Figure 5). In the predicted binding complex, VTPGLQY interacted with TYR51, TRP59, ASN66, ILE88, HIS91, THR92, LYS 118, TRP357, TYR360, HIS387, PHE391, VAL399, ARG402, GLU403, HIS513, PRO519, PROG522, and TYR523. QFPVGR was predicted to interact with ALA170, SER284, ASN285, THR302, VAL373, ASN374, LEU375, and GLU376, and QLQFPVGR was predicted to interact with ASN70, GLU123, HIS353, ALA354, SER355, ALA356, TRP357, HUS383, GLU384, HIS387, PRO407, HIS410, GLU411, PHE457, HIS513, SER517, ARG522, TYR523, and GLY2000 residues.

ACE comprises three major active site pockets that interact with residues: the S1 pocket (ALA354, GLU384, and TYR 523), S1’ pocket (GLU162), and S2 pocket (GLN281, HIS353, LYS511, HIS513, and TYR520) [50]. HIS383, HIS387, and GLU411 in ACE and their binding to zinc ions play a significant role in protein activity by coordinating tetrahedrons [51]. The sites of interaction between these residues are shown in Appendix A. QLQFPVGR forms hydrogen bonds with ALA354, GLU384, and TYR523, which are in the S1 pocket, and express an attractive charge with HIS353 and HIS513, which are in the S2 pocket. VTPGLQY forms a hydrogen bond with TYR523, which is in the S1 pocket, and shows that HIS513, in the S2 pocket, is an attractive charge. VTPGLQY forms a hydrogen bond with HIS387, and QLQFPVGR forms a pi-alkyl bond with HIS383 and hydrogen bonds with HIS387 and GLU411, where the residues interact with zinc ions (Figure 5A,B). The HIS513 and TYR523 residues interacting with VTPGLQY and the residues interacting with HIS353, ALA354, GLU411, HIS513, and TYR523, and those interacting with QLQFPVGR, are included in the residues comprising the complex binding of lisinopril, which is used as an ACE target inhibitor [52]. Moreover, it interacts with VTPGLQY, HIS513, and TRY523 residues, and QLQFPVGR interacts with HIS353, ALA354, HIS513, and TYR523 residues, and provides evidence for strong activity involved in predictive binding to its ACE-inhibition compound captopril [53]. The predicted ACE-lisinopril and captopril binding complexes are shown in Appendix A. These results suggest that these bioactive peptides directly interact with the active site in the S1 and S2 pockets and can influence ACE activity by interfering with zinc-binding motifs [54].

Hydrogen bonds play an important role as catalytic cavities for the stability of ACE and peptides in the complex [55]. QLQFPVGR, VTPGLQY, and QFPVGR interacted with the ACE complex through 19, 18, and 8 binding sites, of which 12 residues (ASN70, GLU123, ALA354, SER355, GLU384, HIS387, HIS410, GLU411, SER517, ARG522, TYR523, and GLY2000), 10 residues (TYR51, ASN66, THR92, LYS118, TRP357, HIS387, VAL399, GLU403, PRO519, and TYR523), and 6 residues (SER284, ASN285, THR302, VAL373, ASN374, and GLU376) were hydrogen bonds, respectively. The relatively weak non-covalent bond, which allows for its transformation into different forms of chains on all protein surfaces, can act as a binding site for peptides of different structures [56]. Unlike the other peptides, QFPVGR formed hydrogen and pi-alkyl bonds at the ACE surface site (Figure 5C). It is speculated that it inhibits activity by preventing interactions at the internal acting site, as this binding complex was formed to surround the surface of ACE. The role of the binding forms in inhibiting the activity by interfering with the interactions between other structures warrants further investigation.

## 3. Materials and Methods

### 3.1. Materials

Abalones were purchased from a fishing village market on Jeju Island. AV was obtained as the shells and muscles were removed. The AV was washed with tap water, and dried for 72 h at 40 °C using far-infrared drying equipment. The dried AV was ground and stored in a freezer at −20 °C. Trypsin were purchased from Sigma-Aldrich (St. Louis, MO, USA). Peptides, identified based on amino acid sequences, were synthesized by Anygen Co., Ltd. (Gwangju, Republic of Korea). All the other chemicals were of analytical grades.

### 3.2. Preparation of Trypsin-Enzymatic Hydrolysate of AV

Enzymatic hydrolysis was performed by slightly modifying the method described in a previous study [57]. Briefly, 50 g of AV powder was blended with 500 mL of distilled water, and the pH was adjusted to 8.0. Trypsin was then added to the mixture at a 1:100 ratio of substrate to enzyme. Then, the reaction was performed for 24 h in a shaking incubator set at 37 °C. At the end of the reaction, the hydrolysate was inactivated at 100 °C for 10 min. The hydrolysate was clarified via centrifugation, filtered, and dried via lyophilization to obtain trypsin-hydrolysate of AV (TAV).

### 3.3. Compositional Analysis of TAV

Protein content was measured using a bicinchoninic acid (BCA) assay with bovine serum albumin as the reference standard [58]. Total polysaccharide content was determined using glucose as a standard in the phenol-sulfuric acid method, as described by DuBois et al., with modifications [59]. The total polyphenolic content was determined using the Folin–Ciocalteu method with gallic acid as a reference standard, as described by Chandler et al., with slight modifications [60].

### 3.4. Purification of Peptides from TAV

The active properties of TAV were isolated using a slight modification of a previous study [61]. The hydrolysate was dissolved in distilled water and loaded onto a Sephadex G-10 gel filtration column (2.5 × 100 cm) equilibrated with distilled water. The elution was performed in each 10 mL tube at a flow rate of 1.2 mL/min, and the elution peaks measured at 220 nm were freeze-dried for the next experiment.

### 3.5. H_2_O_2_ Radical Scavenging Activity

H_2_O_2_ radical scavenging activity was measured as previously described, with slight modifications [62]. Briefly, 20 μL hydrolysate solution at different concentrations was mixed with 100 μL of 0.1 M phosphate buffer (pH 5.0), and 20 μL of 10 mM hydrogen peroxide (in deionized water) was added to a 96-well plate and incubated at 37 °C for 5 min. After incubation, 1.25 mM ABTS of 30 μL and 30 μL (1 unit/mL) of peroxidase were added and kept at 37 °C for 10 min. The absorbance was read at 405 nm using a microplate reader.

### 3.6. Sequencing of Amino Acid from Separated Active Properties

The purified peptides from the TAV were separated on a Zorbax Eclipse Plus C18 column using ultimate 3000 ultra-high-performance liquid chromatography (UHPLC, Thermo Fisher Scientific, Waltham, MA, USA). The masses and amino acid sequences of the separated peaks were identified using quadruple time-of-flight mass spectrometry (Micro Q-TOF III mass spectrometer; Bruker Daltonics, Bremen, Germany) coupled with electrospray ionization (ESI). Molecular masses were determined with a single-charge ([M + H]^+^) or doubly charged ([M + 2H]^+^) positive states in the mass spectra obtained from the isolated peaks. Peptides were automatically selected for fragmentation and their amino acid sequences were confirmed using tandem mass spectrometry analysis.

### 3.7. ACE Inhibition Activity Assay

ACE inhibitory activity was determined as previously described [63]. Inhibition was measured using an ACE kit-WST (Dojindo Inc., Kumamoto, Japan), a colorimetric method. Briefly, the inhibitory effect was determined using enzyme activity and the amount of 3-hydroxybutyric acid made from 3-hydroxybutyryl-Gly-Gly-Gly.

### 3.8. Molecular Docking by Computer Simulation

#### 3.8.1. Preparation of ACE 3D Structure

For molecular docking analysis, the crystal structure of ACE (ID:1O86) was obtained from the Protein Data Bank (PDB). To predict the interactions between the bioactive peptides of interest and the protein, the ACE crystal structure was prepared for docking with the “prepare protein” protocol from the Discovery Studio 2024 (Biovia, San Diego, CA, USA) tool by removing all water molecules, inhibitor lisinopril, and free glycine molecules [64,65].

#### 3.8.2. Preparation of Peptide 3D Structure

Active peptides were manually prepared using Discovery Studio 2024. The molecular dynamics (MD) simulations were performed to select the structures by reducing the absence of solvation during docking and alleviating the influence of packing effects [66]. MD simulation for each step was performed as previously described, with modifications [67]. Briefly, the solvation process was performed using the explicit peripheral of the cubic box cell shape of the peptide, in which the “prepare protein” protocol was followed. Next, simulation was performed with the steepest descent algorithm set to a maximum number of 1000 steps and a root mean squared (RMS) gradient of 1, and the first minimization process was performed, and then the second minimization process was performed with a max number of 2000 steps and an RMS gradient of 0.1 using an adopted-basis Newton–Raphson algorithm. Then, 20 ns of MD was conducted under standard particle number, pressure, and temperature conditions for the peptides, with a 300 ps heating process at 300 K, and the structure of the stable pose was obtained from the final simulation step.

#### 3.8.3. Docking Analysis

The docking of bioactive peptides and ACE was performed using the ZDOCK algorithm to confirm the clustering of poses and binding actions [68]. To allow rapid clustering, the root mean square division (RMSD) and interface cutoff were set to 3° and 6°, respectively. The maximum number of clusters and poses were limited to 30 and 2000, respectively.

### 3.9. Statistical Analysis

Quantitative data were represented as the mean ± standard deviation from triplicate determinations. Statistical comparisons of mean values were performed using one-way analysis of variance (ANOVA) followed by Tukey’s multiple comparison test. Differences were considered statistically significant at *p* < 0.05 (* *p* < 0.05, ** *p* < 0.01).

## 4. Conclusions

In this study, the potential efficacy of fishery by-products was revealed by verifying the antioxidant effects of functional peptides derived from AV and demonstrating the dual effects of simultaneous ACE inhibition activity. Trypsin hydrolysis and the purification of GFC yielded nine bioactive peptides, the H_2_O_2_ scavenging activities of which were measured. The peptide IC_50_ values of LGPY, VTPGLQY, LGEW, LDW, and NLGEW were 0.213, 0.297, 0.289, 0.363, and 0.303 mg/mL, respectively, which showed distinct antioxidant efficacy. The bioactive peptides with dual efficacy showed anti-hypertensive activity, as demonstrated using ACE-inhibitory efficacy assessments in in vitro and in silico analysis. Peptide IC_50_ values of the ACE inhibition of VAR, NYER, VTPGLQY, QFPVGR, LGEW, QLQFPVGR, and NLGEW were 0.104, 0.107, 0.023, 0.023, 0.165, 0.004, and 0.146 mg/mL, respectively, and had low concentrate values, indicating anti-hypertensive effects. QLQFPVGR, VTPGLQY, and QFPVGR displayed superior ZRANK scores based on the docking analysis of ACE inhibition. QLQFPVGR, VTPGLQY, and QFPVGR displayed superior ZRANK scores based on the docking analysis of ACE inhibition. VTPGLQY had a superior dual efficacy in both antioxidant and anti-hypertensive activities, which offers a specialized potential for use as a functional material with both effects. Our results suggest that both LGEW and NLGEW peptides have distinct dual efficacies. In conclusion, TAV, a by-product of fisheries, is a potential bioactive resource with antioxidant and anti-hypertensive properties. Further studies should establish mass-production processes and evaluate their efficacy for industrial applications.

## Figures and Tables

**Figure 1 marinedrugs-22-00461-f001:**
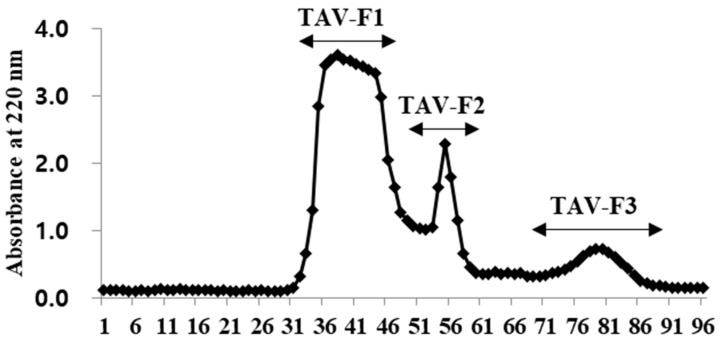
Separation chromatogram of TAV by gel filtration chromatography on Sephadex G-10 Column.

**Figure 2 marinedrugs-22-00461-f002:**
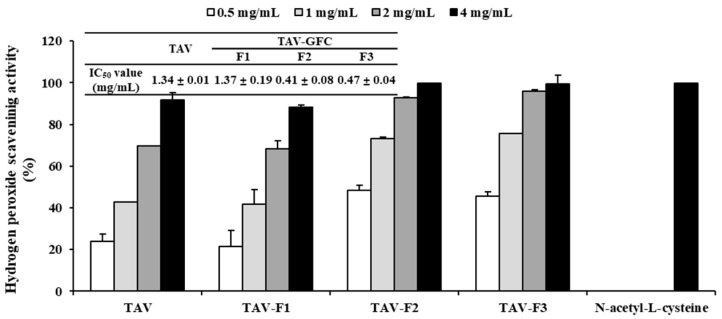
Determination of H_2_O_2_ scavenging activity of TAV and gel filtration chromatography fractions with IC_50_ Values.

**Figure 3 marinedrugs-22-00461-f003:**
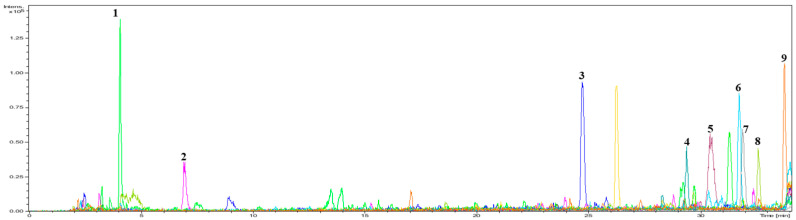
UHPLC analysis chromatogram by C_18_ column obtained from TAV-F2.

**Figure 4 marinedrugs-22-00461-f004:**
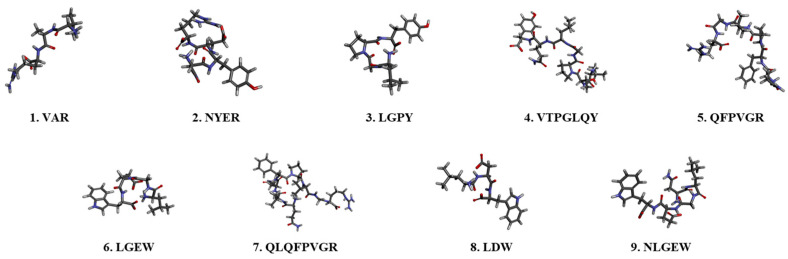
Sequencing results and 3D structures of bioactive peptides derived from TAV-F2.

**Figure 5 marinedrugs-22-00461-f005:**
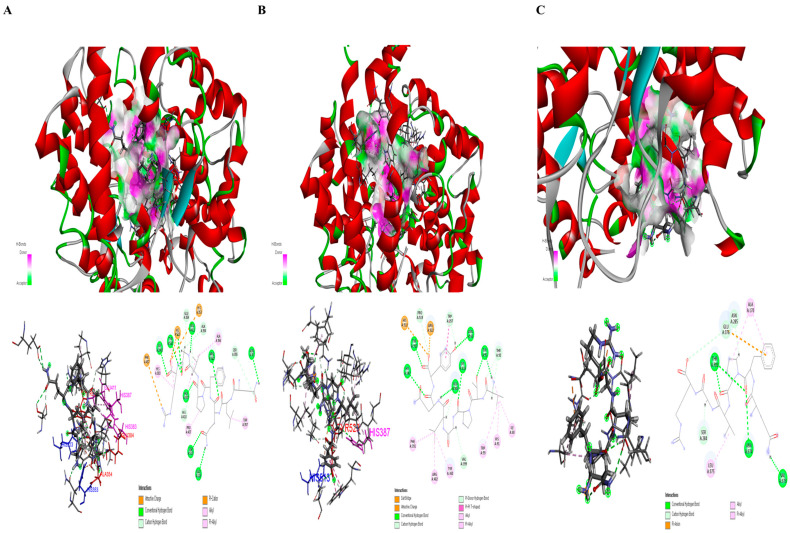
The prediction of the binding site of ACE-bioactive peptide complexes. The ACE-bioactive peptide complexes are shown by favorable hydrogen bond interactions at specific points and represented as stick models of residues with amino acid names and 2D diagrams of the binding complex. QLQFPVGR (**A**), VTPGLQY (**B**), and QFPVGR (**C**) are shown. The amino acid interaction in the active site pockets S1 and S2 are marked in red and blue, respectively, while the amino acid interaction in the zinc ion binding is marked in pink.

**Table 1 marinedrugs-22-00461-t001:** Proximate compositions of TAV.

Proximate Compositions	TAV
Yield (%)	75.44 ± 1.22
Protein (%)	48.19 ± 0.58
Polysaccharide (%)	6.30 ± 0.26
Total phenolic (%)	2.36 ± 0.03

**Table 2 marinedrugs-22-00461-t002:** Determination of mass-to-charge ratio using LC-MS/MS and H_2_O_2_ scavenging activity IC_50_ values of bioactive peptides derived from TAV-F2.

Peak No.	Sequence	Charge	Mass-to-Charge Ratio (*m*/*z*)	H_2_O_2_ Radical Scavenging Activity IC_50_ Value (mg/mL)
1	VAR	1	345.52	N.D ^1^
2	NYER	1	581.27	>0.4
3	LGPY	1	449.24	0.213 ± 0.010 ^a^
4	VTPGLQY	1	777.42	0.297 ± 0.016 ^b^
5	QFPVGR	1	686.37	N.D ^1^
6	LGEW	1	504.25	0.289 ± 0.012 ^b^
7	QLQFPVGR	2	944.54	N.D ^1^
8	LDW	1	433.21	0.363 ± 0.020 ^c^
9	NLGEW	1	618.29	0.303 ± 0.001 ^b^

^1^ Not detected. ^a–c^ Means within a row with different letters are significantly different (*p* < 0.05).

**Table 3 marinedrugs-22-00461-t003:** In vitro and in silico results of ACE inhibitory activity of bioactive peptides derived from TAV-F2.

Peak No.	Sequence	ACE Inhibitory Activity IC_50_ Value (mg/mL)	ZRANK Score
1	VAR	0.104 ± 0.010 ^c^	–49.313
2	NYER	0.107 ± 0.004 ^c^	–53.375
3	LGPY	>1	–55.751
4	VTPGLQY	0.023 ± 0.001 ^b^	–81.500
5	QFPVGR	0.023 ± 0.003 ^b^	–78.016
6	LGEW	0.165 ± 0.011 ^d^	–54.136
7	QLQFPVGR	0.004 ± 0.001 ^a^	–82.468
8	LDW	>1	–53.574
9	NLGEW	0.146 ± 0.009 ^d^	–62.130
-	Captopril	<0.0025 (μM)	-

^a–d^ Means within a row with different letters are significantly different (*p* < 0.05).

## Data Availability

The original contributions presented in this study are included in the article; further inquiries can be directed to the corresponding author.

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
