# Peer review of "The Antioxidant Effects of Trypsin-Hydrolysate Derived from Abalone Viscera and Fishery By-Products, and the Angiotensin-I Converting Enzyme (ACE) Inhibitory Activity of Its Purified Bioactive Peptides"

_marinedrugs, 2024, doi:10.3390/md22100461_

Round 1

Reviewer 1 Report

Comments and Suggestions for Authors

In this study, Heo et al. investigated the antioxidant and ACE inhibitory activities of bioactive peptides derived from abalone viscera. The results indicated that these peptides exhibit significant dual antioxidant and anti-hypertensive effects, making them potential functional materials for health applications. The study is well-structured and clear, and the manuscript is well-written. There are only a few points that need further clarification. Here are some comments on this study:

1.        Lines 42 “affects 30–45% of the population”, it is recommended that the authors state the year when the data were obtained.

2.        Lines 45-49, it is recommended that authors provide some specific examples.

3.        Introduction section, after describing the previous studies (line 62), what were the gaps or shortcomings that caused the researchers to undertake this study?

4.        Line 68, please define the abbreviation “TAV”.

5.        Line 92 “mg/ml”, should be “mL”.

6.        Section 3.2, “the pH was adjusted to 7.0”, the optimum pH for trypsin is around 8 according to Sigma, can the authors explain the reason for setting the pH at 7?

Reviewer 2 Report

Comments and Suggestions for Authors

Dear Editor, Dear Authors,

I evaluated the manuscript « Antioxidant effect of trypsin-hydrolysate derived from abalone viscera, fishery by-product and angiotensin-I converting enzyme (ACE) inhibitory activity of its purified bioactive peptides » by Jun-Ho Heo et al.

In this manuscript, the authors investigated the biological activities of a trypsin-hydrolysate of abalone viscera corresponding to fishery by-product. They focused the screening on antioxidant and ACE inhibition. Nine bioactive peptides (VAR, NYER, LGPY, VTPGLQY, QFPVGR, LGEW, QLQFPVGR, LDW and NLGEW) were purified from digesta. According to data, LGPY, VTPGLQY, LGEW, LDW, and NLGEW exhibited antioxidant properties with IC50 values around 0.2-0.4 mg/mL. Peptides VAR, NYER, VTPGLQY, QFPVGR, LGEW, QLQFPVGR, and NLGEW were found by the authors to inhibit ACE with IC50 values as low as 0.004 mg/mL, respectively. Authors further explored this ACE inhibition by performing docking analysis with the ZDCOK server and found that VTPGLQT interacted with HIS513 and TYR523, and QLQFPVGR interacted with HIS353, ALA354, GLU384, HIS513, and TYR523, contributing to the inhibition of ACE activity. Other peptides were found to interact with amino acids that contribute to stability by binding to zinc ions or with ACE surface sites, suggesting indirect inhibition. Authors conclusions are that hydrolysat of AV is a potential source of bioactive peptides with dual antioxidant and anti-hypertensive dual effects.

I found the study well conducted.

Please find below some comments :

-The H2O2 scavenging activity requires a positive control of the measured activity. In addition, it seems all peptides have almost the same efficiency. It is questionning about the selectivity of the observed effect. I will suggest to test another peptide, not related to AV hydrolysate to see if all peptides, whatever their sequences are active as H2O2 scavenging or not.

-        The ACE inhibition assay needs also a positive control of inhibition by a known inhibitor (ideally a known peptidic inhibitor already used to inhibit ACE) for comparison

-        Table 3 : please correct for peaks 3 and 8 « 1< » to « >1 »

regards

Comments on the Quality of English Language

Dear Editor, Dear Authors,

I evaluated the manuscript « Antioxidant effect of trypsin-hydrolysate derived from abalone viscera, fishery by-product and angiotensin-I converting enzyme (ACE) inhibitory activity of its purified bioactive peptides » by Jun-Ho Heo et al.

In this manuscript, the authors investigated the biological activities of a trypsin-hydrolysate of abalone viscera corresponding to fishery by-product. They focused the screening on antioxidant and ACE inhibition. Nine bioactive peptides (VAR, NYER, LGPY, VTPGLQY, QFPVGR, LGEW, QLQFPVGR, LDW and NLGEW) were purified from digesta. According to data, LGPY, VTPGLQY, LGEW, LDW, and NLGEW exhibited antioxidant properties with IC50 values around 0.2-0.4 mg/mL. Peptides VAR, NYER, VTPGLQY, QFPVGR, LGEW, QLQFPVGR, and NLGEW were found by the authors to inhibit ACE with IC50 values as low as 0.004 mg/mL, respectively. Authors further explored this ACE inhibition by performing docking analysis with the ZDCOK server and found that VTPGLQT interacted with HIS513 and TYR523, and QLQFPVGR interacted with HIS353, ALA354, GLU384, HIS513, and TYR523, contributing to the inhibition of ACE activity. Other peptides were found to interact with amino acids that contribute to stability by binding to zinc ions or with ACE surface sites, suggesting indirect inhibition. Authors conclusions are that hydrolysat of AV is a potential source of bioactive peptides with dual antioxidant and anti-hypertensive dual effects.

I found the study well conducted.

Please find below some comments :

-The H2O2 scavenging activity requires a positive control of the measured activity. In addition, it seems all peptides have almost the same efficiency. It is questionning about the selectivity of the observed effect. I will suggest to test another peptide, not related to AV hydrolysate to see if all peptides, whatever their sequences are active as H2O2 scavenging or not.

-        The ACE inhibition assay needs also a positive control of inhibition by a known inhibitor (ideally a known peptidic inhibitor already used to inhibit ACE) for comparison

-        Table 3 : please correct for peaks 3 and 8 « 1< » to « >1 »

regards

Round 2

Reviewer 2 Report

Comments and Suggestions for Authors

Dear Editor, Dear Authors,

The authors have addressed all my comments.

Regards